# Graphene Derivatives in Biopolymer-Based Composites for Food Packaging Applications

**DOI:** 10.3390/nano10102077

**Published:** 2020-10-21

**Authors:** Ana Barra, Jéssica D. C. Santos, Mariana R. F. Silva, Cláudia Nunes, Eduardo Ruiz-Hitzky, Idalina Gonçalves, Selçuk Yildirim, Paula Ferreira, Paula A. A. P. Marques

**Affiliations:** 1Department of Materials and Ceramic Engineering, CICECO—Aveiro Institute of Materials, University of Aveiro, 3810-193 Aveiro, Portugal; abarra@ua.pt (A.B.); jessica.santos2604@ua.pt (J.D.C.S.); mrfs@ua.pt (M.R.F.S.); 2Department of Chemistry, CICECO—Aveiro Institute of Materials, University of Aveiro, 3810-193 Aveiro, Portugal; claudianunes@ua.pt (C.N.); idalina@ua.pt (I.G.); 3Materials Science Institute of Madrid, CSIC, c/Sor Juana Inés de la Cruz 3, 28049 Madrid, Spain; eduardo@icmm.csic.es; 4Institute of Food and Beverage Innovation, Zurich University of Applied Sciences, 8820 Wädenswil, Switzerland; selcuk.yildirim@zhaw.ch; 5Department of Mechanical Engineering, TEMA—Centre for Mechanical Technology and Automation, University of Aveiro, 3810-193 Aveiro, Portugal

**Keywords:** bionanocomposites, polysaccharides, proteins, polyesters, graphene derivatives, graphene oxide, electrical conductivity, pulsed electric field, food packaging

## Abstract

This review aims to showcase the current use of graphene derivatives, graphene-based nanomaterials in particular, in biopolymer-based composites for food packaging applications. A brief introduction regarding the valuable attributes of available and emergent bioplastic materials is made so that their contributions to the packaging field can be understood. Furthermore, their drawbacks are also disclosed to highlight the benefits that graphene derivatives can bring to bio-based formulations, from physicochemical to mechanical, barrier, and functional properties as antioxidant activity or electrical conductivity. The reported improvements in biopolymer-based composites carried out by graphene derivatives in the last three years are discussed, pointing to their potential for innovative food packaging applications such as electrically conductive food packaging.

## 1. Introduction

Non-sustainable food production and consumption leads to many serious health and environmental problems. It is estimated that one third of the total foodstuffs produced worldwide, 1.3 billion tons, equivalent to one trillion dollars, is wasted. The United Nations (UN) 2030 sustainable development agenda addresses this problem with Goal 12—Ensure sustainable consumption and production [1]. To mitigate this global problem, the development of food processing technologies and packaging materials is of the utmost importance. Packaging offers mechanical and barrier protection against light, dust, pests, gases, moisture, volatiles, and both chemical and microbiological contamination, extending the foodstuffs’ shelf lives while ensuring their quality, safety, and authenticity without requiring additives. However, the sustainability of conventional packaging has become a major issue. The use and disposal of non-biodegradable polymers led to environmental issues across numerous ecosystems [2]. In this context, biopolymers with biodegradable properties have been explored as an alternative to design novel packaging materials that can extend the foodstuffs’ shelf lives and, concomitantly, contribute to minimize the packaging negative environmental impact [3]. Although packaging is a top priority in research, with the development of bio-based and biodegradable plastics representing 53% share of this global demand, their introduction in the food packaging industry is well below expectations [4]. See, for instance, that sustainable packaging market alone is expected to grow from ca. US$ 225 billion in 2018 to over US$ 310 billion by 2024 [5]. The high cost and difficulty to overcome specific technical issues (e.g., viscosity, thermal stability, hydrophilicity, low mechanical strength, and poor gas barrier properties), essential to preserve food quality and safety, are the main obstacles for their large-scale use [6]. Investments in research have been enforced to develop biodegradable food packaging materials, fitting practical criteria. Recently, the incorporation of graphene derivatives into biopolymer has garnered attention due to their potential to provide enhanced mechanical and barrier properties [7,8,9]. This review examines the advances made in the last three years in the field of biopolymer-based biocomposites containing graphene derivatives. Additionally, the most used biopolymers for packaging materials were reviewed. Nowadays, the current design of biocomposites containing graphene derivatives goes far beyond the reinforcement of barrier and mechanical properties. Some graphene derivatives, such as reduced graphene oxide (rGO) with radical scavenging capacity, may assign an antioxidant activity to the packaging materials, an extremely important property to enhance foodstuffs’ shelf lives [10]. An interesting breakthrough introduced through use of graphene derivatives is electrical conductivity, which is advantageous for the development of food packaging materials suitable for their application in pulsed electric field technology (PEF) [10,11,12].

## 2. Biopolymers as Food Packaging Raw Materials

Biodegradable biopolymers have been gaining societal, scientific, industrial, and economic importance to develop sustainable food packaging bioplastics, alternatives to conventional non-biodegradable fossil-fuel-based plastics [6,13]. In recent years, several approaches to develop biodegradable plastic materials from biomass feedstock by biocatalytic transformation, chemical synthesis, or simple feedstock polymers’ extraction have received much attention, taking advantage of using renewable resources as raw materials during polymer production/purification. This strategy contributes to a sustainable essence of the final packaging while advancing toward a circular economy [6]. According to nova-Institute data [4], simultaneous bio-based and biodegradable polymers had a 33% increase in applications within the plastic packaging industry between 2017 and 2019, particularly dominated by starch-based blends, polylactic acid (PLA), and polyhydroxyalkanoates (PHA) polymers (Figure 1). However, the number of scientific research articles focused on the development of bio-based and biodegradable formulations for food packaging application has had an even more notable increase, 78%, during the last three years. In this section, the available and emergent bio-based and biodegradable polymers under research, namely polysaccharides, proteins, and polyesters, presented in Figure 1, will be discussed as well as their production sources, technical properties, and challenges within the food packaging industry.

### 2.1. Polysaccharides

#### 2.1.1. Starch

Starch is constituted by two distinct polymers, the mostly linear amylose, α-d-(1,4)-linked glucopyranosyl units, and the highly branched amylopectin, α-d-(1,4)-linked glucopyranosyl units partially substituted by α-d-(1,6) linkages units (Figure 2), with a proportion of 20–30% and 70–80%, respectively, depending on its botanical source [13,14,15]. It is considered one of the most promising natural biopolymers since it is easy to acquire, it can be extracted from foodstuffs byproducts (has four conventional sources such as wheat, corn, potato, cassava, and other sources including fruit waste [16]), and it is biodegradable and low-cost [17]. It is important to note that more than 310 tons of starch are industrially produced from corn, potato, or wheat for packaging purposes, accounting for approximately 25% of all used bio-based and/or biodegradable plastics (Figure 1) [4].

Starch has been used to form edible or biodegradable films and granulates through casting from aqueous solutions, film blowing, forming, injection molding, blow molding, and extrusion [18,19,20,21,22].

Plasticizers, along with temperature (for gelatinization) [16], allow the increase of the starch-based films flexibility, improving processability. Starch-based films have reasonable gas barrier properties; however, they still present some sensitivity to humidity conditions. Moreover, they also exhibit poor moisture barrier and mechanical properties, limiting their applications as packaging materials [23,24]. Furthermore, starch retrogradation naturally increases the matrix crystallinity over time, promoting brittle films [25,26], another factor that prevents the starch-based materials’ widespread adoption. Nevertheless, starch has been used as a filler in plastics, allowing to reinforce their structure [27,28]. Within commercial perspectives, plasticized starch is often blended with a range of other polymers, like polyethylene (PE), polypropylene (PP), polystyrene (PS), Poly(3-hydroxybutyrate-co-3-hydroxyvalerate) (PHBV), or PLA in a proportion range between 30% and 70% [29], where biodegradability of the final product is dependent on the degradation abilities of the polymeric compounds [13]. There is a huge interest and new research focusing on the trade-off between biodegradability and functionality of starch-based plastics, promoting their efficiency and versatility. These blended materials have been considered as an alternative to pristine PE, PP, or PS-based plastics [4,30,31].

In a laboratorial environment, other interesting future scalable approaches have been explored to improve starch-based materials. Gonçalves et al. [16] used oil and waxes recovered from potato frying residues and potato peels, respectively, to tailor the surface properties (roughness and wettability) and flexibility of starch-based films. Chollakup et al. [32] found out that the incorporation of compounds such as cinnamon oil and fruit peel extract can provide antibacterial and antioxidant properties. Still, in both examples, the final product is not suitable for applications as food packaging at industrial scale. Currently, the best option is, as mentioned, to use starch blended with other more resistant materials (PE, PP, PLA, etc.).

#### 2.1.2. Cellulose

Cellulose is a linear polymer of several hundred to many thousands of (β1→4) linked D-glucose units (Figure 2); it is the most abundant natural biopolymer on Earth [33], it is biodegradable, and it is commercially derived by a delignification process from wood pulp, which contains 40–50% of cellulose by weight [34]. This biopolymer, by itself, is unsuitable for film production due to its high crystallinity and long fibers [33,35], which make it have a non-thermoplastic nature (it cannot be softened or melted by the application of heat, nor can it be processed [36]) due to its strong intra- and inter-chain hydrogen bonds [37]. Chemically modified forms of cellulose fare a better chance against water, while exhibiting good mechanical properties, such as cellophane, cellulose esters (cellulose nitrate and cellulose acetate), and cellulose ethers (carboxymethyl cellulose and hydroxyethyl cellulose) [38].

A notable material derived from acid hydrolysis of cellulose is the nanocrystalline cellulose (CNC). It has been widely explored in numerous areas in food packaging (Figure 1) [39,40,41]. CNC has high elastic modulus, optical transparency, low thermal expansion coefficient, good gas barrier properties, and low toxicity, thus being biocompatible and biodegradable [42]. In packaging, CNC has been used as a gas barrier and filler of nanocomposites [39,40,43]. Microcrystalline cellulose (MCC) and nanofibrillated cellulose (NFC) have also used been for similar purposes [44,45]. Nanocellulose-based films are usually combined with other biopolymers, as chitosan and PHA, or plasticizers (e.g., glycerol, sorbitol, methoxypolyethylene glycol (MPEG) [46]) to improve or modify their physicochemical properties and extend their application range [47,48,49,50]. These biocomposites have shown excellent mechanical and oxygen barrier properties; however, their performances rapidly decline in the presence of moisture [49,50,51,52]. To overcome this drawback, nanocellulose grafting or blending with hydrophobic compounds as tannins, cholesterol, lignin, and fatty acids have been investigated [51,53,54]. However, these solutions often showed insufficient improvements on the hydrophobicity performance and thus are still unsuitable for using as food packaging materials.

#### 2.1.3. Chitosan

Chitosan (CS) is a linear polysaccharide of 2-amido-2-deoxy-β-d-glucoses attached by (β1→4) linkages (Figure 2), obtained by the deacetylation of more than 50% of chitin [55,56]. Chitosan’s molecular weight depends on its source and, on average, can vary from 50 to 1000 kDa [57]. Manipulation of deacetylation percentages allows tailoring of its physicochemical properties and degradation profile [58]. Studies comparing low, medium, and high CS molecular weights found the latter molecular weight (>300 KDa) achieved better results in packaging [59,60]. CS is only soluble in acidic media, and its chains become positively charged when the pH level is under their pKa ≈ 6.5 [61,62]. CS also contains a primary amino group (NH_2_) that can be protonated to NH_3_^+^ and readily form electrostatic interactions with anionic groups in an acid environment. This characteristic allows the incorporation of a variety of chemical groups [63] (e.g., grafting phthalic anhydride, which increases antibacterial properties against Gram-positive and Gram-negative bacteria [64]) and letting prepared systems react to external stimuli such as temperature [65].

This biopolymer has been widely studied due to its potential in areas such as material science [66] and pharmacology [67], since it can be obtained at low cost, has large scale availability, is nontoxic, and is biodegradable [58]. Moreover, the antioxidant, antifungal, and antibacterial activities of CS has been attracting special attention in the food packaging sector as demonstrated in Figure 1 [68,69]. Additionally, because CS also has selective permeability to CO_2_ and O_2_, it has been investigated as edible coatings for fruit packaging applications. For this particular use, it delays the rate of respiration, decreases both weight loss and ethylene production, inhibits postharvest diseases, and increases the antioxidant process, allowing to extend the product shelf life while preserving the fruits overall quality. These results were achieved just by using CS as a packaging solution [17]. When applied to films formation, CS gives rise to transparent, flexible, and good oxygen barriers [17,70]. However, CS-based materials have some limitations such as low water vapor barrier characteristics and low mechanical strength; thus, they still require additives as fillers or plasticizers (e.g., glycerol [71]) to overcome these fragilities for food packaging applications [72,73].

#### 2.1.4. Alginate

Alginate is the general name given to the family of linear polysaccharides consisting of binary copolymers, made up of (1–4) linked β-d-mannuronic acid (M) and α-l-guluronic acid (G) monomers (Figure 2), occurring in different proportions and distributions across the chain, depending on the source [74]. Thus, the molecular weight of alginate can vary between 32 and 400 kDa [75]. Alginate is usually extracted from brown algae, mainly *Laminaria hyperborean*, *Macrocystis pyrifera*, and *Ascophyllum nodosum*. In addition, microbial alginate can be produced by *Azotobacter vinelandii* and *Pseudomonas aeruginosa* [76]. The alginate ability to form strong gels/low-soluble polymeric materials in the presence of divalent cations, commonly Ca^2+^, due to the formation of a three-dimensional (3D) structure designed as an “eggbox” model, has been explored for the production of biodegradable and edible films [76]. The main advantages of alginate are: chemical stability, controllable swelling properties, and low content of toxic, pyrogenic, and immunogenic contaminants [74]. Alginate has been studied for new packaging options in the form of casted films [71,77] and as food-grade edible coatings [78]. Modified alginate such as propylene glycol alginate [79], sodium alginate [80,81], or calcium alginate [82] are frequently applied. The brittleness and low moisture barrier properties are the main disadvantages of alginate materials, which can be optimized with the use of diverse additives. Plasticizers such as glycerol and sorbitol improved the alginate films’ flexibility [71]. The combination of alginate with other biopolymers such as PLA can increase the tensile strength and the oxygen barrier properties [80]. In the same way, blends with biopolymers, e.g., soy protein, increased the tensile strength and decreased the water vapor permeability and water solubility [79]. Inorganic additives are also employed to enhance the performance of alginate materials. Sulfur nanoparticles with antimicrobial activity improved the mechanical properties of alginate films while increasing the UV barrier properties and hydrophobicity [83]. Similarly, alginate composites containing antimicrobial silver nanoparticles improved the shelf lives of fruits and vegetables [81].

#### 2.1.5. Pullulans

Currently, the term ‘‘pullulan” is used in the literature to mean not only the ‘‘polymaltotriose”, the maltotriose -(α1à4)Glc*p*-(α1à4)Glc*p*-(α1à6)Glc*p* trimer (Figure 2), produced by different strains of fungus-like yeast *Aureobasidium* spp.- *Aureobasidium pullulans*, *Aureobasidium melanogenum*, and *Aureobasidium mousonni*, [84,85,86] but also other slightly different polysaccharide varieties (e.g., aubasidan-like, or pullulan-like), still similar to the pullulan, produced by distinct *Aureobasidium pullulans* varieties (e.g., *Aureobasidium pullulans var. aubasidani* and *Aureobasidium pullulans var. pullulans,* each with numerous distinct strains) [87].

This microbial extracellular polysaccharide is a biodegradable, biocompatible, non-mutagenic, nontoxic, hygroscopic (meaning that depending on the relative humidity at which it stored it can absorb water), non-carcinogenic, and edible polymer [88,89]. To create pullulan films or coatings, numerous techniques such as solvent casting [90,91], extrusion [92,93], coating by dipping or spraying [94,95], layer-by-layer assembly [96,97] or electrospinning [98,99,100] can be used. The resulting films are devoid of color, opacity, taste, and odor, and are also heat-stable and impermeable to both oil and oxygen [88,89]. Still, these films present drawbacks such as poor mechanical properties, namely their brittleness and their inability to resist water due to their hydrophilic nature and lack of active functions [88]. Thus, it is necessary to combine pullulans with other polymers and/or plasticizers [98,101,102] or even (nano)particles, which act as fillers [88,103,104], to improve the materials barrier properties (water and oxygen) and improve mechanical properties (e.g., tensile strength). The use of other materials such as silver particles can also impart new properties such as antimicrobial activity [103,105].

### 2.2. Proteins

#### 2.2.1. Gelatin

Gelatin is derived from the fibrous insoluble protein called collagen (by thermal denaturation of collagen in the presence of diluted acid [106]) and is typically obtained from bones, skin, and connective tissue generated as waste during animal slaughtering and processing [107]. It is a heterogeneous mixture of single- or multi-stranded polypeptides, each with extended left-handed proline helix conformations and containing between 300 and 4000 amino acids most of which make up glycine, proline, and 4-hydroxyproline residues. Its structure is a mixture of α-chains (one polymer/single chain), β-chains (two α-chains covalently crosslinked), and γ-chains (three covalently crosslinked α-chains) [108]. Gelatin’s typical amino acid composition is Ala-Gly-Pro-Arg-Gy-Glu-4Hyp-Gly-Pro- (Figure 2) [107]. There are two types of gelatin of animal origin: Type A, with an isoelectronic point at pH ~8–9, obtained from acid treated collagen; and Type B, with an isoelectronic point at pH ~4–5, derived from an alkali-treated precursor which converts asparagine and glutamine residues into their respective acids, resulting in higher viscosity. The gelatin derived from pig skin is normally type A and the one from beef skin or pig cattle hides and bones is type B [109].

Gelatin is abundant, low-cost, and biodegradable [106]. It has been broadly studied in food packaging (Figure 1); however, it is hygroscopic, and consequently, it tends to swell or to be dissolved when put in contact with foodstuffs with high moisture content [106]. Another major disadvantage is its poor mechanical properties, especially when wet (poor water vapor permeability).

To overcome these issues, several approaches have been explored, namely addition of different molecules such as crosslinkers (to reduce solubility), plasticizers, and antimicrobial or antioxidant compounds [110,111]; combining gelatin with other biopolymers such as starch [112,113], chitosan [114,115], and whey protein [116], among others, to improve mechanical and water resistance properties; and adding nanoparticles such as silver, for instance, which can impart antimicrobial properties [114,116]. Recently, gelatin has been explored to create coatings and films for fruit packaging purposes due to its excellent ability to form films, its good oxygen barrier capacity, and UV-absorbing properties mediated by the presence of aromatic amino acids in its structure [117,118,119,120,121].

#### 2.2.2. Zein

Zein, seen in Figure 2, is a natural prolamin which can be separated by sodium dodecyl sulfate polyacrylamide gel electrophoresis (SDS-PAGE) into α-, β-, γ-, and δ-zein [122]. It is the main storage protein of corn at 45–50%, it is water insoluble (due to the hydrophobic character of its apolar amino acids, proline and glutamine [123]), and it is nontoxic and biodegradable [124,125].

Zein is an option for food packaging because it has excellent oxygen and carbon dioxide barrier properties and high thermal resistance [123,126]. Furthermore, in comparison with films produced from other proteins, it presents higher tensile strength and lower water vapor permeability [123]. However, films produced solely from zein present fragilities specifically related with brittleness, low surface functionality, and poor mechanical properties (such as lack of flexibility [127]) needed for industrial processing. Additionally, they cannot resist a high relative humidity condition. All these factors limit their use in food packaging applications [127].

To improve zein films, several different options have been explored: adding plasticizers (e.g., oleic acid, polyethylene glycol, glycerol, etc.) [128,129], adding micro- or nanoparticles (e.g., zinc or silver) [129,130], using other biopolymers to create better biopolymer blends (e.g., PP, PLA, etc.), [123,125,129,131,132] or adding other compounds (e.g., tannic acid, gamma-Cyclodextrin, etc.) which can provide different supplementary abilities (e.g., antioxidant, antibacterial properties) [130,132,133].

### 2.3. Polyesters

#### 2.3.1. Polylactic Acid

Lactic acid is a monomer produced by fermentation of carbohydrates by bacteria, mainly *Lactobacillus* which by polymerization originates the polylactic acid (PLA—Figure 2), a thermoplastic aliphatic polyester [134]. PLA has good mechanical and thermal properties [17] which are strongly related to the ratio between the two mesoforms D and L [17]. PLA has high stiffness, strength, and water and oxygen permeability levels comparable to polystyrene [17,135]. PLA has excellent properties, namely high transparency, rigidity, and biodegradability, and it can be produced from renewable sources [37,136]. It can also tolerate various types of processing technologies, namely injection molding, extrusion, blow molding, and thermoforming [137]. However, PLA also has several shortcomings such as poor heat resistance, brittleness, poor melting strength, low degradation rate, and a narrow processing window [138,139,140]. To overcome these drawbacks, PLA-based products are being designed not only using PLA but also by mixing it with other biodegradable (bio)polymers and non-biodegradable resins and/or by compounding PLA with fillers such as fibers or micro- and nanoparticles [141,142,143]. Currently, PLA is being used in packaging applications as films, as thermoformed blow-molded containers and as short shelf life bottles [139,140,142,143]. Commercially, it is a good candidate to replace PE, PS, and polyethylene terephthalate (PET), which are non-biodegradable polymers commonly used for food packaging [37]. More than 160 tons of PLA are annually produced at the commercial level for plastic packaging purposes, which represent about 13% of all bio-based and/or biodegradable materials, making it the second most used biodegradable polymer (after starch-based blends) within this sector [4].

#### 2.3.2. Poly(3-hydroxybutyrate-co-3-hydroxyvalerate)

Poly(3-hydroxybutyrate-co-3-hydroxyvalerate) (PHBV), seen in Figure 2, is an aliphatic polyester from the PHA family, known by its biodegradability, nontoxicity, and biocompatibility [144]. PHA are linear polyesters that naturally occur in a variety of microorganisms and are accumulated intracellularly as carbon and energy reserves [13]. Currently, this natural ability of some microorganisms has been explored in the industrial synthesis of these biodegradable polymers by fermentation using renewable matrices as carbon sources [13,38]. Within the packaging sector, 96 tons of PHA were produced in 2019 from sugarcane pulp, sugar beet pulp, corn, potato, and wheat [29]. The majority of PHA are composed of 3-hydroxy fatty acid monomers [13]. A huge variety of PHA exist, and this diversity allows the production of PHA with a wide range of properties to compete with conventional plastic materials applied for food packaging [38]. Some PHA exhibit similar thermal and mechanical properties to PE, PS, and PP, which make them promising for food packaging applications [13,37,38]. However, this process is still not very cost effective and so the use of PHBV is limited due to its high production cost [144]. PHBV consists of a poly(3-hydroxybutyrate) or PHB with a few added 3-hydroxyvalerate (HV) units [145]. Moreover, PHBV has great oxygen barrier properties, chemical inactivity, high viscosity in a liquid state (good for extrusion processes), and has better mechanical properties, namely increased surface tension and more flexibility compared to PHB [144,145]. Nonetheless, PHBV still has some considerable deficits, namely brittleness, a narrow processing window, poor thermal and mechanical properties, no antimicrobial properties, and low resistance to water vapor permeability [145,146]. To improve these properties, fillers as clays, cellulose nanocrystals, and metal oxides [146,147,148,149] have been successfully used, enhancing thermal stability, mechanical properties, and barrier properties [146,147,148,149].

## 3. Graphene Derivatives-Based Biocomposites as Food Packaging Materials

As mentioned in the previous section, biopolymers by themselves do not present all the necessary requirements intended for food packaging, namely in terms of mechanical resistance and water/gas barrier properties. Furthermore, there is an ongoing trend towards active and intelligent packaging, i.e., approaching packaging that is able to interact with food to actively extend its shelf life, and to inform the consumer about the safety state of the food products, respectively [150]. In this context, graphene derivatives may play a role on assigning activity and intelligence to the packaging, in addition to other properties.

### 3.1. Graphene Derivatives

Graphene is one of the most promising nanomaterials which is constituted by a flat monolayer of carbon atoms in a two-dimensional (2D) hexagonal lattice, held together by a backbone of overlapping sp^2^ hybrids bonds [151]. This structure confers to graphene many remarkable properties, such as the strongest mechanical robustness with a modulus of over 1060 GPa [152], without losing its molding properties, large surface area [153], impermeability to gases [154], and optical transparency [155], among others. Graphene was considered by the scientists A.K. Geim and K.S. Novoselov, who isolated it for the first time, to be the mother of all graphitic forms because it is the 2D building material for carbon structures of all other dimensionalities, and because it can be encased into 0D fullerenes, folded into 1D nanotubes, or stacked into 3D graphite [151]. When short stacks of graphene sheets are packed, having a platelet shape, they receive the name of graphene nanoplatelets (GNP) [156]. Graphene oxide (GO) is a derivative of graphene obtained from graphite in two steps. The first step is the chemical oxidation (with strong acids) of graphite into graphite oxide, followed by simple stirring or mild sonication (mechanical process) to exfoliate graphite oxide into single layers [157]. Thus, GO is a carbon layer with several oxygen functional groups (carboxylic, hydroxyl, and epoxy) on its basal planes and at its edges, resulting in a hybrid structure of sp^2^ and sp^3^ configurations [158]. The high oxygen content has been demonstrated to be very useful for the chemical modification/functionalization with other molecules, thus allowing its dispersion in different matrices enabling the preparation of nanocomposites with interesting properties [159,160]. Globally, GO retains plenty of the properties of the graphene, but it is much easier and cheaper to prepare in bulk quantities and to process (better dispersion in different solvents). In fact, the chemical exfoliation of graphite in oxidative medium originates stable aqueous suspensions of GO. Depending on the practical application of GO, a reduced form can be more suitable—for example, to be incorporated in a polymeric matrix of hydrophobic nature [161]. Thus, GO can be reduced (rGO) by chemical, thermal, microwave, photo-chemical, and photo-thermal or microbial/bacterial methods, where the material can recover partial or complete hybridized sp^2^ configuration, thus approaching the graphene configuration [162,163,164].

Interestingly, graphene derivatives have been widely used as polymer reinforcement and are well known to impact on several properties of the final nanocomposite, namely on its mechanical, thermal, electrical, conductive, and fire retardancy properties, to name just a few [150,151,152,153,154,155,156]. Due to the already described variety of graphene derivatives, the term “graphene” is often used in a generic manner to describe indifferently each of these nanostructures, thus creating misinterpretations about its properties [165]. Owing to the chemical and structural features presented by each graphene derivative, they have different properties that make them proper additives for specific applications. The most fundamental properties of graphene derivatives to consider are: (i) number of graphene layers, (ii) average lateral size, and (iii) oxygen content (with a variable carbon-to-oxygen (C/O) atomic ratio). The scheme presented in Figure 3 helps to visualize the categorization of different graphene derivatives types according to the three fundamental properties mentioned previously [165].

The use of graphene derivatives in biopolymer-based composites for food packaging applications is still lightly explored. The Web of Science search of the topic “biopolymers food packaging composites” displays 234 results, but when refined including the word “graphene”, only nine articles are displayed (accessed on 22 September 2020). However, given the potential of these nanomaterials to improve determining properties of biopolymeric composites for food packaging application, this will certainly be an expanding area with high impact in the future. Hereafter, some of the most relevant properties obtained for biopolymer-based composites including graphene derivatives will be reviewed.

### 3.2. Properties of Biopolymer-Based Composites with Graphene Derivatives

The first step in the development of graphene-based biopolymer composites is to guarantee the uniform dispersion of graphene materials into a polymer matrix, which might not be easy given the high propensity for self-agglomeration of graphene-based materials, as a result of the strong van der Waals forces and *π*–*π* electrostatic interactions between nanosheets or nanotubes, and/or their common weak dissolvability in water and organic solvents during biocomposite fabrication [152]. The solution mixing is considered the most common and simplest method to prepare polymer composites. In this method, fillers are mixed with the polymer solution, homogenized by physical stirring, and the composites recovered after solvent evaporation [166]. However, achieving a good dispersion is still a challenge. The low dispersion of these nanofillers and the consequent weak interfacial adhesion between the graphene-based filler and the host biopolymer can compromise the final morphological, mechanical, barrier, and physicochemical properties of the films [167]. Sonication, ultrasonication, high-speed blending, and melt blending are some strategies used to achieve a proper dispersion of these nanofillers in the bio-based matrices [168,169,170]. When these methodologies are not enough to achieve a good dispersion, the wrapping of surfactants using noncovalent interaction methods [171] or the chemical modification of graphene materials by covalent bonding [145,167] are some of the approaches used. Recently, as an alternative to chemical modification, oil-in-water Pickering emulsions were used to disperse carbon nanotubes (CNT) into PLA matrix using cellulose nanocrystals as stabilizer and as dispersant due to its amphipathic character [172]. After the dispersion, films are commonly obtained by solution casting method [7,170] or by melt processing, which is a more convenient method for the potential industrial application [173,174,175]. The final structural organization of these composites containing graphene derivative-based fillers (originally suggested for layered silicates) is categorized in three types, namely phase separated, intercalated, and exfoliated (Figure 4). The performance of composites is maximized in their exfoliated form, corresponding to the carbon nanostructures well dispersed within the matrix. This condition provides a tortuous diffusion barrier and creates a good percolation network, which greatly enhances the overall properties [176].

#### 3.2.1. Mechanical and Thermal Stability Properties

The preparation of biopolymers nanocomposites with GNP [152,173,178,179], chemically modified rGO [145,167,180,181], or CNT [172,174,175,182] reinforcement has exhibited significant enhancements in the mechanical properties of chitosan, PLA, alginate, PHBV, cellulose nanofibers, and starch even at very low concentrations (<5%), as described in Table 1. The Young’s modulus, tensile strength, and elongation at break are increased with the incorporation of the diverse graphene derivatives as fillers. For example, compared with the neat biopolymer, the incorporation of only 0.7 wt% rGO into a PHBV-based formulation increased these properties by 100%, 119% and 24%, respectively [145]. In turn, the incorporation of 2 wt% of CNT into a PLA matrix also provides an improvement of 52% and 36% on the materials’ tensile strength and elongation at break, respectively [7]. There is, therefore, a reinforcement of mechanical properties by the incorporation of graphene derivatives, and the different morphology of these nanostructures does not seem to have a predominant or limiting role for this mechanical improvement [183]. In some cases, the incorporation of graphene derivatives into biopolymer-based formulations could result in a slight decrease of elongation at break, meaning that these composites can present lower flexibility [152,170,178]. However, this drawback can be overcome or limited through the synergism of graphene derivatives with other nanofillers, such as cellulose nanocrystals (CNC) or ZnO [173,178]. The simultaneous use of different graphene derivatives as fillers could also be a strategy to enhance the mechanical performance of final composites since the reinforcement of PLA-based films with a 0.4 wt% of a hybrid co-filler with added single-walled carbon nanotubes (SWCNT) and GO resulted in an increased tensile strength and Young’s modulus up to 75% and 130%, respectively [7]. Mechanical properties of biocomposites are directly related to the biopolymer(s) interaction type established with the carbon filler(s), such as electrostatic interactions, van der Waals forces, or hydrogen bonding between the hydrophilic groups of the biopolymer(s) and the oxygenated functional groups on carbon nanostructures, or chemical bonding when grafting processes are applied as illustrated on Figure 4. This interaction can improve the stress transfer mechanism between graphene derivatives and the biopolymer matrix, thus increasing the material rigidity and tensile strength of the nanocomposite [145,167]. In the case of GO or chemically modified GO, mechanical properties are, therefore, strongly influenced by the degree oxidation [152], since the incorporation of the same amount of GO (0.6 wt%) prepared with growing ratios of KnMO4/graphite (from 2:1 to 8:1) into a CS matrix generated variable mechanical properties: Young’s modulus decreasing and increasing by −20% to 307%, tensile strength increasing by 15% to 327% and elongation at break increasing and decreasing by +1% to −29% with the oxidation degree increase when compared with this neat biopolymer. The increase of the oxidation degree led to an increase of the rigidity and strength, accompanied by a stretchability decrease, which is linked to the chain mobility restriction as a result of more hydrogen bonds forming. The oxidation degree of rGO is, thus, a factor that should be taken into account on the optimization of biocomposites processability. In addition, the grafting of rGO with cellulose nanocrystals (rGO-CNC) potentiates the formation of hydrogen bonds between biopolymer/graphene derivative, and thus better mechanical results are achieved even when compared with the simultaneous incorporation of non-grafted rGO and CNC [145]. The combined use of rGO and CNT seems to potentiate the Young’s modulus and tensile strength increase [184]. Generally, the upper reinforcement degree of rGO or CNT and their chemically modified forms is usually reflected on the improvement of tensile strength and Young’s modulus. However, at high concentrations (>20 wt%), graphene derivatives can introduce defects in the biocomposite microstructure, compromising the further improvement of mechanical properties [179,183]. In addition, it is still necessary to assess the mechanical performance of graphene-based biocomposites as food packaging relative to shelf life. Food products emit compounds such as CO_2_ which, in the presence of water, form carbonic acid that can interact with the biocomposites and influence their resistance [185].

The thermal stability of the biocomposites can also be improved with the inclusion of graphene derivatives into biopolymer-based formulations. The temperature of degradation is delayed by 5 °C up to 20 °C, when compared with the biopolymer film without these nanofillers [145,167]. The high heat resistance of graphene derivatives seems to improve the thermal stability of biocomposites since the introduction of these nanofillers into biopolymer matrix creates an effective heat and gas barrier which prevents the diffusion of volatile degradation products and radicals, slowing the biocomposite thermal degradation [145,167].

#### 3.2.2. Barrier Properties

Graphene-based biocomposites exhibit great barrier properties against gases, water vapor, and UV light [7,145,152,167,173,179]. The gas barrier properties have a special importance on modified atmosphere in food packaging. Water vapor barrier properties should avoid the water penetration on packaging atmosphere or the dehydration of foodstuffs, and UV light barrier should avoid the transmittance of UV light across packaging materials and the subsequent degradation of organic compounds present in the food matrices, thus preventing its deterioration and quality lost. A hybrid co-filler made with SWCNT and GO fillers with a loading of only 0.4% into PLA films decreased the oxygen transmission rate by 67% and diminished the transmission of ultraviolet-visible light by 30%. Manikandan et al. [8] proved that the use of polyhydroxybutyrate biocomposite containing 0.7 wt% GNPs had improved water, oxygen, and UV barrier properties and was also able to promote a fourfold increase of potato chips and milk products shelf life. The improvement in gas and water barrier of graphene-based biocomposites is explained by the introduction of impermeable graphene-based nanofillers with large surface area which restricts the motions of the biopolymer chains, and the creation of a tortuous pathway into the film matrix that complicates gas molecules diffusion.

#### 3.2.3. Surface Hydrophobicity Properties

Surface hydrophobicity of biopolymer-based materials is also enhanced by the incorporation of graphene derivatives. As described in the Section 2, the hydrophilic character of the bio-based and biodegradable polymers is the main limiting factor for their application. In contact with food, these biopolymers can absorb water or even solubilize and, concomitantly, compromise their protective function as packaging films. The integration of graphene derivatives into biocomposites increases their surface hydrophobicity and decreases their water solubility. Recently, the use of 0.6 wt% of rGO grafted with maleic anhydride and subsequently with dodecyl amine increased the contact angle of PLA/starch composite from 67° to 81° [167]. Furthermore, solubility of CS in water decreased from 38% to 22% with rGO at 0.5 wt% [152]. These values are promising, but they are not enough for these composites’ application as food packaging films.

#### 3.2.4. Biodegradability

The development and optimization of graphene-based biocomposites is ongoing and it is important to prove that the biodegradable properties of natural polymers are not compromised. The biodegradation of polymers occurs as a result of the activity of specific microorganisms to hydrolyze and oxidize these molecules in a short time after their disposal. The strong interaction between graphene fillers and the biopolymer can improve the mechanical stability, as previously mentioned; however, the degradation rate can be slightly reduced by delaying the diffusion of water into the polymeric matrix [189]. Lyn et al. [152] supported this hypothesis by demonstrating that solvent casting CS/rGO composites biodegrade completely after 20 days of composting. On the contrary, melt processing films of PLA/starch blends with 5 wt% of rGO have 17–26% higher weight loss than the blends without the rGO in 183 days of controlled aerobic composting process [190]. Nonetheless, despite the degradation rate decreasing, the biocomposites biodegradability is not compromised [152,189,190].

#### 3.2.5. Active Properties

Nowadays, in addition to the desirable biodegradable packaging, there are global research interests for the development of active food packaging to extend shelf life, enhance safety, and maintain the organoleptic properties. Biocomposites with graphene derivatives can exhibit antioxidant, antimicrobial, and antifungal activity as extensively reviewed by Carvalho et al. [191]. The radical scavenging capacity of rGO can provide antioxidant activity to biocomposites. CS-based films with 20–33 wt% of rGO showed an increase of inhibition in the range of 54% to 82% after 8 h of incubation [170] and this activity can avoid the oxidation of packaged foodstuffs. The antimicrobial and antifungal properties of graphene-based nanostructures are based on their capacity to induce cell membrane disruption and oxidative stress that compromise bacterial proliferation and sporulation [191]. Biocomposites with graphene derivatives revealed in vitro antibacterial properties against a broad spectrum of pathogenic microorganisms (such as *Enterococcus faecalis*, *Staphylococcus epidermidis*, *Escherichia coli*, *Staphylococcus aureus*, *Staphylococcus haemolyticus*, and *Bacillus subtilis)*. The surface modification of graphene derivatives with essential oils or other metal compounds such as Ag, ZnO, or TiO_2_ has been adopted as strategy to enhance the inherent antimicrobial potential of these nanostructures [191]. Recently, a PLA/CNT/cinnamaldehyde film revealed great potential application as a controlled-release antibacterial active food packaging film with an active effect proven up to 21 days [192]. Antifungal activities of biocomposites containing carbon nanostructures against *Aspergillus niger*, *Cryptococcus neoformans*, *Candida tropicalis*, *Candida albicans*, *Botrytis cinereas*, and *Rhizopus spp.* have also been described [191].

Alternatively, the delocalized conjugated electron structure of graphene derivatives has been recently explored for production of photocatalytic hybrid systems with application in organic compounds’ degradation [186,193]. Organic compounds’ degradation might have special importance for controlling the composition inside of the modified atmosphere packaging in order to extend the shelf life of food products. The organic compounds’ degradation ability of visible light-responsive GO/Bi_2_WO_6_ hybrid system was shown to be 4.4 times more effective on ethylene degradation than pure Bi_2_WO_6_ when incorporated into starch-based films [193]. Ethylene is a phytohormone responsible for inducing the fruit ripening process and its accumulation inside packaging leads to senescence of fruit. The application of starch/GO/Bi_2_WO_6_ composites as active packaging materials of highly perishable fruit can be an opportunity to decrease fruit losses and wastes [193].

Therefore, the use of graphene derivatives into biopolymer-based composites reveals to be a promising approach to develop active food packaging, enhancing the preservation of food products.

#### 3.2.6. Clay–Graphene Bionanocomposites

Nano- and micro-particulated clay minerals could be incorporated as fillers leading to biopolymer–clay nanocomposites (bionanocomposites) based on the assembly of polysaccharides, proteins, polyesters, etc., with layered silicates, such as smectites and vermiculites, as well as fibrous clays like sepiolite and palygorskite [194,195,196,197,198,199,200,201]. These clay-based biocomposites are of great interest for diverse applications, including films and coatings for use in food packaging, as they can improve barrier and mechanical properties, in addition to conferring water resistance and other characteristics very attractive for these types of uses [200].

Clays could be assembled to carbon particles producing graphene-like materials of variable composition and properties, including adsorption ability and electrical conductivity, mainly afforded by the clay component and the graphene derivative component, respectively. These carbon–clay materials could be synthetized following two main processing approaches: (i) top-down, by crushing together both components, and (ii) bottom-up, by growing carbon on clay minerals used as supports [202,203]. Another interesting preparative way to develop carbon–clay composites is based in the use of the fibrous clay known as sepiolite, which is processed and commercialized as a rheological grade product (Pangel^®^). It is well known that the dispersion of carbon NP—for instance, carbon nanotubes—in aqueous medium [204,205,206,207,208] is of paramount importance to homogeneously disperse carbon NP as fillers in water soluble biopolymers. An efficient approach for this purpose is based on the use of the rheological grade sepiolite assisted by ultrasonic irradiation in the presence of those carbon NP leading to dispersions stable for several months without syneresis effects [209]. For instance, the stabilization of MWCNT and GNP was initially interpreted in terms of steric stabilization where sepiolite nanofibers act as interposed species in between the carbon NP avoiding its reassembly [209]. Based on this behavior, diverse films of carbon–clay bionanocomposites containing gelatin, polyvinyl alcohol and alginate provided with electrical conductivity have been reported [210].

Clay–graphene materials dispersed in diverse biopolymers are fillers of great interest because they can introduce significant improvement on the resulting biocomposites, increasing their barrier properties and conferring electrical conductivity. In fact, synergistic effects of clay minerals and graphene-based materials are observed for the use of clay–graphene systems as fillers of polymers. For instance, GNP/sepiolite can be used as fillers of alginate leading to films showing simultaneously reinforcing properties together marked in-plane conductivity (ca. 500 S m^−1^), which increases by the additional incorporation of MWCNT favoring percolation reaching values up to 2500 S m^−1^ [202,210]. These alginate-based biocomposite films show variable mechanical properties (e.g., elongation at break and Young’s modulus) depending on the proportion of the starting components, i.e., sepiolite, GNP, and alginate [210]. A clear disadvantage of the use of sepiolite in the preparation of these carbon–clay fillers can be the possible loss of mechanical properties. Moreover, high clay content significantly reduces the electrical conductivity in detriment of its use as films for pulsed electric field (PEF) application.

### 3.3. Emerging Application for Biopolymer-Based with Graphene Derivatives

The use of electrically conductive graphene derivatives in food packaging can be an important feature for the development of food processing technologies. Food processing by pulsed electric field (PEF) is one of the most promising emerging non-thermal technologies. The PEF treatment consists in the application of short high voltage pulses to food products to inactivate enzymes and microorganisms, while maintaining their sensorial and nutritional properties [211,212,213]. Currently, foodstuffs are processed before packaging in direct contact with the chamber electrodes [214]. This methodology compromises the food safety due to metals releasing from the stainless steel electrodes and post-sterilization recontamination events (before food packaging) [215]. The release of metals from electrodes could be avoided by covering the electrodes with an electrically conductive material, while the recontamination could be prevented if the PEF treatment occurred after packaging (Figure 5). Roodenburg et al. [216] used a commercially available electrically conductive plastic, commonly used to pack electronic components, to cover the chamber electrodes. The use of this composite constituted by ethylene vinyl acetate and 30 wt% carbon black particles, with an electrical conductivity of 0.75 S m^−1^, led to a bacterial inactivation of 2.1 log_10_. The same authors also studied the effectiveness of PEF in-pack, using this composite material as a food packaging pouch. The results showed a bacterial inactivation of 5.9 log_10_, which reaches the pasteurization level [217].

After the PEF proof of concept, it is necessary to develop electrically conductive materials suitable for food packaging. Biopolymer-based composites containing graphene derivatives can be a solution. In this regard, CS-based biocomposites containing rGO were recently suggested for electrically conductive food packaging [170]. In this work, rGO was prepared using caffeic acid as reducing agent, representing a green alternative to toxic reducing agents that should be avoided in food packaging applications. The bionanocomposite films containing 50 wt% rGO showed an electrical conductivity of 0.7 S m^−1^ and 2.1 × 10^−5^ S m^−1^, in-plane and through-plane, respectively. CS-based flexible films containing rGO-Fe_3−x_O_4_ were also suggested for electrically conductive food packaging [12]. These sustainable bionanocomposites, produced in absence of toxic chemicals, achieved an in-plane electrical conductivity of ~0.016 S m^−1^ with 50 wt% rGO-Fe_3−x_O_4_. In both works [12,170], the electrically conductive films also display mechanical properties and antioxidant activity attractive for food packaging applications. However, to the best of the authors knowledge, there are no other reports specifically addressing this issue. Therefore, Table 2 lists reports from the last three years regarding electrically conductive biocomposites suggested for other applications that are considered to have potential for electrically conductive food packaging. The graphene derivatives found as fillers in these biocomposites were MWCNT, SWCNT, rGO, GNP, and graphene. PLA, cellulose, and CS are the most used biopolymer matrices. Polymer blends are a strategy to improve the mechanical properties of the matrix and the dispersion of carbonaceous fillers [190,218]. Films containing GNP, as conductive filler, and a matrix of a blend of PLA biopolymer and poly (butylene adipate-co-butylene terephthalate) (PBAT) synthetic biodegradable polymer were successfully prepared [219]. PLA has a weak affinity with GNP, while PBAT has a good affinity thus enabling to hold high GNP loadings. This strategy created conductive channels by confining GNP into the PBAT continuous phase and constructed good percolation networks leading to an electrical conductivity of 338 S m^−1^. The MWCNT were the most used fillers due to their high electrical conductivity capacity, typically required for electronic applications [172,174,175,201,218,220]. MWCNT were also used is small amounts as doping materials to enhance the electrical conductivity of GNP in the CS matrix [221]. GNP and rGO are cost-effective alternatives given the large-scale production of food packaging [222]. However, the rGO with high electrical conductivity is usually prepared using toxic reducing agents that prevent its use for food packaging. On the other hand, the use of nontoxic reducing agents produces rGO with lower electrical conductivity [12,170,190,223,224]. The in-situ reduction of GO into NFC with hydroiodic acid (HI) lead to an in-plane electrical conductivity of 22.22 S m^−1^, while the same procedure using ascorbic acid lead to an electrical conductivity of 0.83 S m^−1^. The films thermally reduced (TR) at 450°C achieved the highest conductivity of 23.42 S m^−1^, but this method weakened the mechanical properties [225].

The food packaging materials for PEF treatment in-pack should have a through-plane electrical conductivity close to the electrical conductivity of the packaged food, which is typically between 0.1–2 S m^−1^ [216]. Many of the biocomposites listed in Table 2 present superior electrical conductivity to the food conductivity. However, these values refer to the in-plane conductivity, since the through-plane conductivity is not reported in most cases. These fillers have a preferential alignment in the plane direction, leading to composites with low through-plane conductivity, even in cases of high in-plane conductivity [170,210]. Thus, the through-plane electrical conductivity of graphene-based nanomaterials is very poor [226]. Therefore, the development of novel strategies to improve the through-plane conductivity are paramount to this emerging application.

**Table 2 nanomaterials-10-02077-t002:** Biopolymer-based composites containing graphene derivatives suitable for electrically conductive food packaging recently described in literature.

Biopolymer	Graphene Derivative	σ (S m^−1^)	Applications	Ref.
**PLA**	4.3 wt% MWCNT	59.30	EMI shielding	[172]
	15 wt% GNP	0.35	-	[227]
	3 wt% MWCNT	6.42	EMI shielding	[175]
	2 wt% MWCNT	19.70	EMI shielding	[174]
	5 wt% SWCNT	1010	Organic devices	[228]
	15 wt% GNP	0.36	Electronics	[229]
**PLA/Starch** **PLA/PBAT** **Cellulose derivatives**	5 wt% rGO	0.001	Packaging	[190]
	40 wt% GNP	338	Electronic devices	[219]
	5 wt% rGO	15,200	Electronic devices	[230]
	4.5 wt% MWCNT	10	Electrochemical devices	[201]
	10 wt% MWCNT	37.6	Electronics	[231]
	9 wt% graphene	2.4	Diverse	[223]
	9 wt% rGO	1.4	Diverse	[223]
	50 wt% rGO (AC)	0.83	Electronic devices	[225]
	50 wt% rGO (HI)	22.22	Electronic devices	[225]
	50 wt% rGO (TR)	23.42	Electronic devices	[225]
**Cellulose/SPI** **CS**	0.25 wt% *MWCNT	0.82	-	[218]
	50 wt% rGO	0.7	Food packaging	[170]
	50 wt% rGO-Fe_3-x_O_4_	0.016	Biomedical	[12]
	55 wt% GNP/5 wt% MWCNT	2900	Bioelectrocatalysis	[221]
	2.5 wt% rGO	0.08	Biomedical	[224]

PLA: polylactic acid. MWCNT: multi-walled carbon nanotubes. EMI: electromagnetic interference. GNP: graphene nanoplatelets. SWCNT: single-walled carbon nanotubes. rGO: reduced graphene oxide. PBAT: poly (butylene adipate-co-butylene terephthalate). AC: ascorbic acid. HI: hydroiodic acid. TR: thermally reduced. SPI: soy protein isolate. CS: chitosan.

## 4. Future Perspectives

The returns of the combination of graphene derivatives with biopolymers were reviewed. The electrical conductivity provided by the graphene derivatives opens the way for the development of novel and auspicious materials and technologies, namely electrically conductive packaging materials for food processing by PEF.

The electrically conductive biocomposites show great potential to be used in intelligent packaging as flexible platforms to incorporate specific sensing molecules since the target detection would also promote changes in the electrical conductivity of the biocomposites. In this context, a PLA-based sensor containing a layer of paper coated with GO was recently described [232]. This sensor monitors the food quality through the detection and quantification of biogenic amines. The PLA biopolymer was used as a matrix to entrap colorimetric sensing compounds that detect the biogenic amines, while the ability of GO to adsorb and desorb organic molecules allowed its quantification by laser desorption–ionization mass spectrometry. This research area is expected to expand in the near future.

The presence of clays assembled to graphene derivatives is very attractive for the preparation of multifunctional nanoplatforms with potential applications in food packaging. As reported above (Section 3.2.6), the barrier properties regarding gases and water of layered graphene derivatives such as GO and GNP could be enhanced by the simultaneous presence of clays—in particular, for smectite silicates like montmorillonite. Carbon–clay materials could be functionalized by the immobilization of semiconducting NP such as TiO_2_ or ZnO on the clay surface [233]. This strategy appears as a promising way to prepare improved UV-shielding films for packaging food, taking into account the high performance of this type of semiconductor as a UV barrier in biopolymer films [234], given that a synergistic effect with the graphene derivatives component present in these composites was expected in our case. Remarkably, these semiconducting NP are also provided with bactericidal activity. In addition, it may be considered that clay minerals are efficient adsorbents for many diverse organic compounds as well as support of metal and metal oxide nanoparticles [235,236,237]. Therefore, the presence of clays in the carbon–clay materials could be of interest to modulate the properties of the resulting films based on clay–graphene biocomposites. For instance, bioactive compounds such as bactericide or antifungal drugs might be easily incorporated into the pores of sepiolite and other clay silicates acting as carriers of these bioactive agents. Moreover, characteristics as scavengers of oxygen can be introduced in the fillers by anchorage of metal oxide NP (e.g., iron-oxides) on the external surfaces of clays.

Prior to the application of these materials, it is important to understand the risks of their migration into food. Until now, there are few literature evidences, and those studies indicate that there is no migration of graphene derivatives into food, or that the migration occurs far below the international migration limits [145,169]. Additionally, these materials also constrain the diffusion and migration of plasticizer molecules for food systems [167]. Still, there is a need to further investigate the risks involved in the use of graphene derivatives in biopolymers. Notably, recent studies sustained by sound science-based assessment of the potential impact on health and environment are being developed with the aim to understand the graphene derivatives properties that control their biological effects with promising results towards their safety application under controlled safe-by-design approaches [238,239,240].

## Figures and Tables

**Figure 1 nanomaterials-10-02077-f001:**
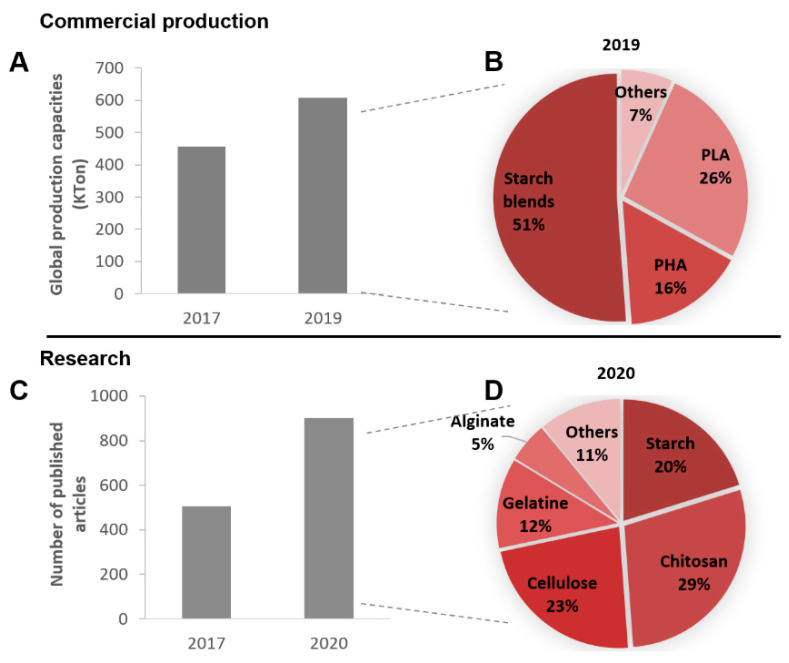
Upper scheme: (**A**) Evolution of commercial production capacities of bio-based and biodegradable polymers within the packaging plastic market between 2017 and 2019; (**B**) the representativity of different biopolymers on the 2019 market. Data for the graphical construction acquired from the European Bioplastics Report [4]. Lower scheme: (**C**) Number of published research articles focused on the various bio-based and biodegradable polymers combined with the term “food packaging” in 2017 and 2020; (**D**) the distribution of the diverse biopolymers in research in 2020 (accessed on 7 October 2020 through “The Web of Science”).

**Figure 2 nanomaterials-10-02077-f002:**
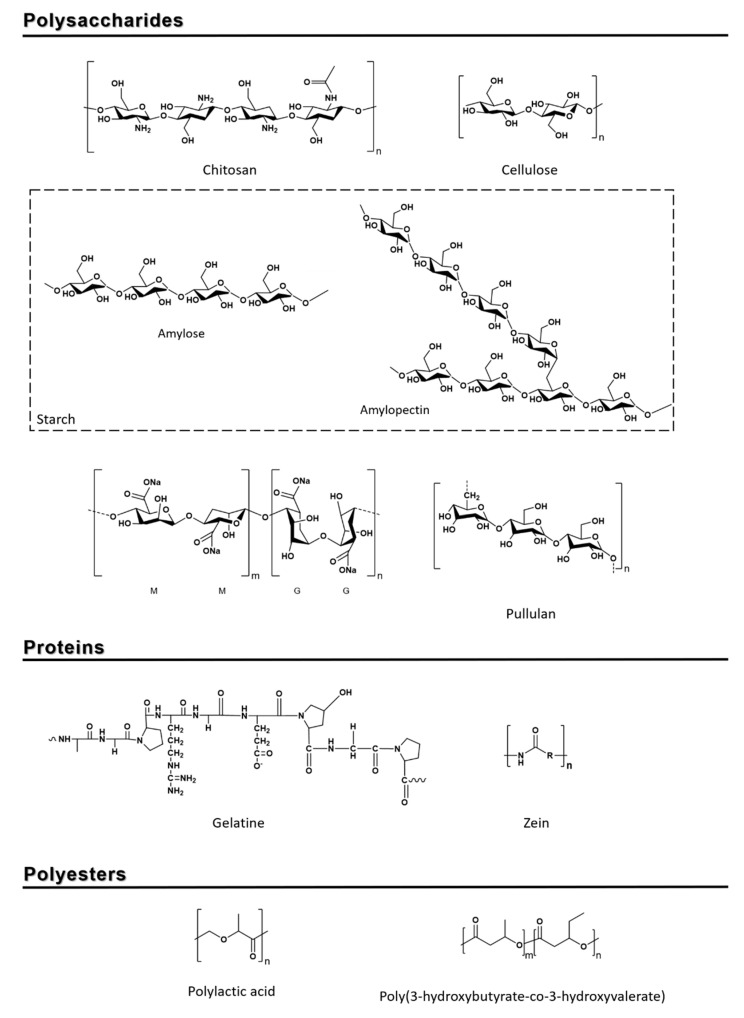
Structures of biopolymers.

**Figure 3 nanomaterials-10-02077-f003:**
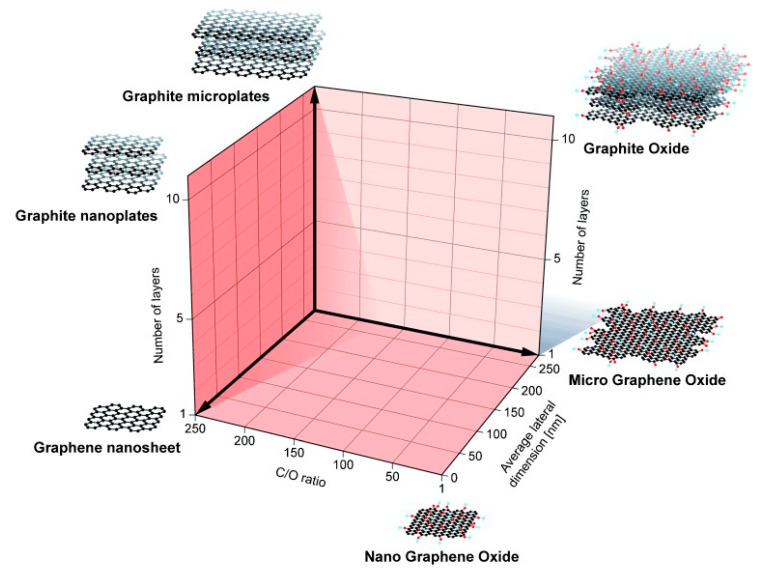
Classification grid for the categorization of different graphene types according to three fundamental graphene derivatives properties. The different materials drawn at the six corners of the box represent the ideal cases according to the lateral dimensions and the number of layers reported in the literature. The values of the three axes are related to the graphene derivatives at the nanoscale, but it is feasible to expand the values to the microscale. Reproduced with permission from [165]. Copyright Wiley-VCH Verlag GmbH & Co, KGaA, Weinheim, Germany, 2014.

**Figure 4 nanomaterials-10-02077-f004:**
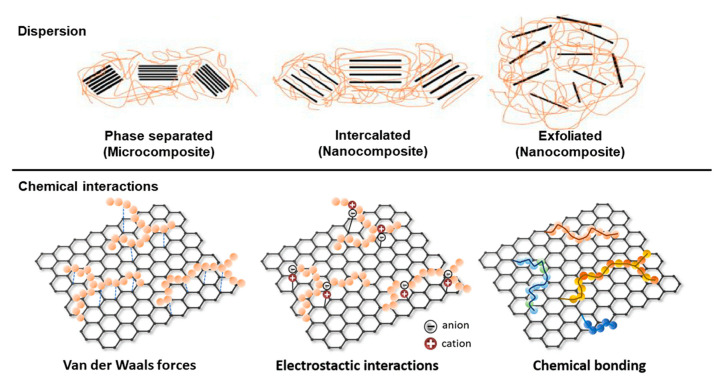
(**Upper scheme**) Dispersion of platelet-like fillers into polymer composites. Adapted from reference [177]. Copyright Elsevier, 2018. (**Lower scheme**) Interactions between typical carbon nanostructures and different polymers in composites.

**Figure 5 nanomaterials-10-02077-f005:**
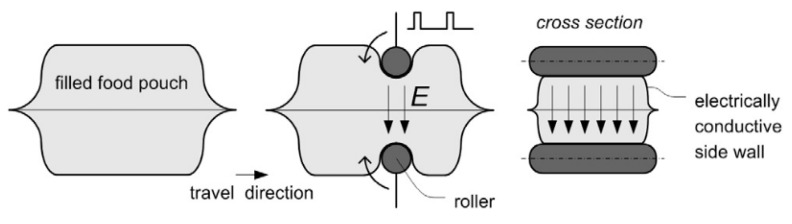
PEF processing in-pack. Food is packaged into an electrically conductive food packaging prior to PEF Table 2. Adapted from reference [217] Copyright Wiley-VCH Verlag GmbH & Co, KGaA, Weinheim, 2011.

**Table 1 nanomaterials-10-02077-t001:** Preparation methods and mechanical properties of biopolymer/carbon nanostructures composites.

Polymer	Nanomaterial	Preparation Method	Nanomaterial Dispersion Strategies	Main Mechanical Effects	Ref.
**PHBV**	0.5 to 0.7 wt% GO, 1:0.5 wt% non-grafted GO/CNC, 1 wt% grafted GO-CNC	Solvent casting	Physical blending (stirring); chemical grafting	Covalently grafted GO-CNC achieved the highest YM, TS, and EB values, which were up to 138%, 170%, and 52% higher than neat polymer.	[149]
**CS**	0.5 wt% GO with different degrees of oxidation	Solvent casting	Ultrasonic dispersion	By increasing of oxidation degree of GO, the TS and YM increase and the EB decreases.	[152]
	0.25 wt% GO and 3 wt% ball-milled maleamic acid–isobutyl polyoctahedral silsesquioxanes (MAIPS)	Solvent casting	Physical blending	Synergistic reinforcements were found on the composite with GO and MAIPS: highest YM and TS (e.g., 50% and 38% higher, respectively, than neat polymer).	[186]
	5 wt% GNP and 5 wt% ZnO	Solvent casting	Ultrasonic dispersion	The simultaneous incorporation of GNP and ZnO lead to highest values of YM and TS, and to a slight decrease of EB.	[178]
	GO (0, 25, 40, 45, 48, or 50 wt%, in relation to CS weight)	Solvent casting	Ultrasonic dispersion	CS/GO showed higher TS (improvements of 70% to 110%), YM (improvements of 500%), and lower EB (decay of 90%) when compared with chitosan films. No significative differences were found in CS-based composites with 40 to 50 wt% of GO.	[170]
	0 to 30 wt% GNP or MWCNT	Solvent casting	Ultrasonic dispersion	At the same ratios, CS/GNP and CS/MWCNT exhibited similar TS and YM values. The highest values of TS were achieved by incorporation of 15 wt% of GNP or MWCNT, which represented improvements of 49% and 64% when compared to those of neat polymer. In turn, the highest values of YM were achieved by incorporation of 30 wt% of GNP or MWCNT, which represented improvements of 109% and 115% when compared to those of neat polymer.	[183]
**Starch**	3, 6, and 9 wt% MWCNT grafted with ascorbic acid (AA-MWCNT)	Solvent casting	Ultrasonic dispersion	The YM and TS were reduced and the EB was increased by enhancing the AA-MWCNT loading in the composite.	[182]
**PLA**	1 wt% GNP and CNC (ratio 50/50)	Hot pressing	Melt blending with the Triton X-100 surfactant	Improvements on YM, TS, and EB were achieved by simultaneous incorporation of both nanofillers.	[173]
	0.5 wt% GO and 1 wt% CNC	Solvent casting	Physical blending	Increase of PLA/CNC/rGO nanocomposite TS up to 23%.	[187]
	0.05 to 2 wt% GO-Ag hybrids	Solvent casting or direct mechanical melt blending	Physical blending or melt blending	Higher flexural strength was achieved when higher amounts of GO-Ag hybrids were added and when physical blending and solvent casting subsequent methods were applied.	[188]
	2 wt% CNT	Solvent casting	Ultrasonic dispersion	The TS and EB have an enhancement of 52% and 36%, respectively, in comparison with PLA films.	[174]
	0.5, 1.0, 2.0, and 3.0 wt% MWCNT	Injection molding	Mechanical blending	Increments of 32.70% and 67.17% were obtained for the TS and EB with the inclusion of 3 wt% of MWCNT.	[175]
**PLA/CNCs**	0.9 to 8.3 wt% CNT	Compression molding	Pickering emulsions	The mechanical performance of the sample was maintained a high level (tensile strength: 45.52 MPa, Young’s modulus: 3152 MPa) after the incorporation of 4.3 wt% CNT.	[172]
**Alginate**	0 to 25 wt% GO	Solvent casting	Physical blending	The inclusion of >2 wt% GO content into alginate-based composites demonstrated remarkable improvements in YM. The maximum upgrade achieved was of 230% in comparison with pure alginate (15 wt% GO). The evolution of the TS suggested the inclusion of defects in the microstructure as GO increased.	[179]

PHBV: poly(3-hydroxybutyrate- co -3-hydroxyvalerate). GO: graphene oxide. CNC: cellulose nanocrystals. YM: Young’s modulus. TS: tensile strength. EB: elongation at break. CS: chitosan. MAIPS: ball-milled maleamic acid–isobutyl polyoctahedral silsesquioxanes. CNT: carbon nanotubes. PLA: polylactic acid. MWCNT: multi-walled carbon nanotubes. GNP: graphene nanoplatelets. ZnO: zinc oxide. Ag: silver particles.

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
