# Peer review of "Graphene Derivatives in Biopolymer-Based Composites for Food Packaging Applications"

_nanomaterials, 2020, doi:10.3390/nano10102077_

Round 1

Reviewer 1 Report

Article ”Graphene derivatives in biopolymer-based 2 composites for food packaging applications” is well a written review about different polymer based composites for food packing. I really appreciated the parts about the disadvantages of each polymer used. I do not know why authors did not show the same criticism in chapters dealing with graphene (wait maybe because of the title).

In most of the application mentioned graphene derivatives are limited to a filler. The question was raised can graphene derivatives penetrate products. If it will be GO than is waters soluble definitely it will penetrate a moist meat or anything that has water inside. If it will be covered/protected by polymers than the role of graphene limits to a filler and there are cheaper fillers than graphene, GO e.g. graphite nanoparticles, carbon blacks etc. For now, graphene is a science trend which will end if the production costs will not decrease.

Authors presented also in the chapter graphene derivatives parameters that are valid for single flakes (I would say as a bite) which are not in reach for real life packages made out of it. Someone people who does not know that will expect marcels from this material.

If that is not too much trouble, you could add to the figure 1 a circular plot of market share of different types of packages to show which of them is used at most, which is the cheapest, has at least disadvantages.

Please add also the missing references in text (7 in total)

In general article is fine, I recommend it for publication after small corrections

Author Response

We recognize the reviewer concerns about possible “graphene derivatives penetration in products”. In fact, and as the reviewer mentioned, this review focus on graphene derivatives mostly used as fillers of the polymeric matrixes. The wt.% of the fillers content in the polymer matrix is usually very low. Also, these fillers must be well dispersed and integrated within the polymer to take advantage of their properties. Thus, It is not expectable that the filler will be released during the package use. However, this situation must be considered and studied, that is why several research groups are dedicating their research to the toxicity assessment of these new products. In section 4 (Future perspectives) we mention this important issue and cited few references reporting studies which evidence that there is no migration of graphene derivatives into food. Still, this is a new field of graphene derivatives application and a lot of research studies need to be performed before they reach the market.

Concerning the price of these fillers, their cost is being reduced during the last years. However, the final price of the package will always depend on its final properties and application. For example, the use of pristine graphene, which may be considered the most expensive one, may extend the use of the package to smart packaging. One of the examples mentioned in our revision is the electrical conductivity, which is advantageous for the development of food packaging materials suitable for their application in pulsed electric field technology (PEF). So, the price is always relative and depend on the final application.

Regarding the properties mentioned for graphene single flakes, in fact we started by presenting (subsection 3.1) the top properties of graphene derivatives, however we were cautious to highlight the enormous variability of graphene derivatives and consequently thir properties, as well illustrated in Figure 2 (now, Figure 3).

Following the reviewer suggestion, we decided to add a new figure (Figure 1), where we introduced: A) chart with the evolution of commercial production capacities of bio-based and biodegradable polymers within packaging plastic market between 2017 and 2019; B) the representativity of different biopolymers on 2019 market. Data for the graphical construction acquired on European Bioplastics Report [14th European bioplastics Conference. Institute for Bioplastics and Biocomposites - Nova Intitut Bioplastics market development update 2019; C) Number of published research articles focused on the various bio-based and biodegradable polymers combined with the term ‘food packaging’ in 2017 and 2020; D) The distribution of the diverse biopolymers in research in 2020 (accessed on the 7th October 2020 through ‘The Web of Science’).

The reference list was updated.

Reviewer 2 Report

The paper is a well written, useful summary of the current status of the use of graphene derivatives in biopolymer-based composites for food packaging applications. As the authors assess, this is a poorly explored topic in the web science and this review encourages to explore this area and underlines the relevance and the impact of the use of graphene derivatives in food packaging formulations, giving important perspectives. All the bio-materials commonly used for food packaging are described in their structure and properties. The limits in the use of these bio-materials, in terms of low mechanical resistance and water/gases barrier properties, are also reported. This manuscript does an excellent job demontrating how graphene derivatives can improve the physicochemical and mechanical properties of the biopolymer-based composites. 

Author Response

We thank the reviewer comments.

Reviewer 3 Report

The manuscript is really interesting, however, there are some suggestions for the authors:

  1. Line 143-163, the authors can provide more information about Chitosan, such as the molecular weight of chitosan, how can apply to food packaging materials?
  2. Line 164-178, the authors can provide more information about Alginate.
  3. Line 287-347, the authors write a lot of words and information about Graphene derivatives, however, there are not enough references to support it. Thence, the authors must improve it.

Author Response

We thank the reviewer for the positive comments and suggestions. We followed these recommendations, and more information was added about chitosan and alginate, in the respective sections.

Regarding the information about Graphene derivatives, this was supported by the inclusion of eight more references (155, 157-162 and 165).